# Numerical Study on an RBF-FD Tangent Plane Based Method for Convection–Diffusion Equations on Anisotropic Evolving Surfaces

**DOI:** 10.3390/e24070857

**Published:** 2022-06-22

**Authors:** Nazakat Adil, Xufeng Xiao, Xinlong Feng

**Affiliations:** College of Mathematics and System Sciences, Xinjiang University, Urumqi 830017, China; nazakatm@stu.xju.edu.cn (N.A.); fxlmath@xju.edu.cn (X.F.)

**Keywords:** convection–diffusion equation, Lagrangian, evolving surface, radial basis function–finite difference, anisotropic radial basis function

## Abstract

In this paper, we present a fully Lagrangian method based on the radial basis function (RBF) finite difference (FD) method for solving convection–diffusion partial differential equations (PDEs) on evolving surfaces. Surface differential operators are discretized by the tangent plane approach using Gaussian RBFs augmented with two-dimensional (2D) polynomials. The main advantage of our method is the simplicity of calculating differentiation weights. Additionally, we couple the method with anisotropic RBFs (ARBFs) to obtain more accurate numerical solutions for the anisotropic growth of surfaces. In the ARBF interpolation, the Euclidean distance is replaced with a suitable metric that matches the anisotropic surface geometry. Therefore, it will lead to a good result on the aspects of stability and accuracy of the RBF-FD method for this type of problem. The performance of this method is shown for various convection–diffusion equations on evolving surfaces, which include the anisotropic growth of surfaces and growth coupled with the solutions of PDEs.

## 1. Introduction

Convection–diffusion equations on stationary or evolving surfaces are becoming increasingly popular since they arise in a wide variety of applications [1,2,3,4]. For PDEs on stationary surfaces, it is a common practice that the smooth surface is discretized as a triangulated surface, i.e., piecewise linear surface. The popular surface finite element methods [5,6] have been used in this context. A lot of work has been done to derive the finite element methods for solving PDEs on evolving surfaces [3,7,8]. In the finite element framework, the mesh evolving approach is used for the discretization of surface evolutions. However, regenerating a global mesh is expensive for large mesh deformations.

Recently, meshfree methods have become popular due to their flexibility of working on solving surface PDEs. The meshfree methods allow large deformations or topological changes of surfaces to be handled more easily than the mesh-based finite element methods. In the meshfree methods, surfaces are only discretized with scattered points. The RBF method is a high accurate numerical approach in the context of meshfree methods. The global RBF collocation method proposed in [9] provides a high order numerical scheme for solving stationary surface PDEs. The global RBF method has a highly computational cost because it leads to a dense discrete matrix. When it comes to PDEs on evolving surfaces, the dense matrix needs to be recalculated at each time layer. Therefore, the local RBF-FD methods provide a competitive numerical discretization for surface PDEs. The RBF-FD methods have been successfully developed to convection–diffusion and reaction–diffusion PDEs on domains [10,11] and surfaces [12,13,14,15]. The RBF-FD methods have also been applied to the evolving surface PDEs [16,17].

In this paper, the surface differential operators are approximated by using the tangent plane method (TPM) in the RBF-FD framework. The TPM was first proposed by Demanet [18]. The key idea of the method is that the surface differential operators are approximated on the tangent space. Recently, the TPM was extended to the context of the meshfree generalized finite difference method and applied to surface PDEs on stationary [19] and evolving surfaces [20]. In [21], the RBF-FD method is combined with TPM to calculate the differential weights for the surface Laplacian. In this paper, we focus on the application of the RBF-FD TPM method on solving evolving surface PDEs. The main advantage of this method comes from the simplicity of the discretization of surface operators.

Furthermore, we found that some anisotropic surface evolutions [3] may introduce the trouble of distortion on the nodes distribution. For example, surfaces evolve in ways that depend linearly on the underlying axes. In this case, the irregular clustering of nodes may lead to results in conditioning problems of the RBF interpolation. To improve the performance of the RBF-FD TPM, we couple the method with ARBF interpolation [22,23,24] to solve this kind of problem.

The rest of the paper is organized as follows. In Section 2, we give the formulation of a convection–diffusion equation on evolving surfaces. In Section 3, we firstly review the basic notation of ARBF interpolation. Then we show how to calculate the differentiation weights by using the ARBFs within the TPM framework. In Section 4, the PDE is discretized in time. In Section 5, we analyze the accuracy and eigenvalue stability of the approximation for the Laplacian on an ellipsoid using the proposed method. In Section 6, various numerical examples are performed to verify the effectiveness of the proposed method for different surface evolutions. Finally, we end with conclusions and some further issues.

## 2. Convection–Diffusion Equation on Evolving Surfaces

In this section, we consider the convection–diffusion equation on a time-dependent closed and smooth submanifold $\Gamma \left(t\right)\subset R^{3}$:(1)$$\frac{d}{dt}u+u\nabla_{\Gamma (t)}\cdot v-k\Delta_{\Gamma (t)}u=f\left(t,x\right),\;\mathrm{on}\;\Gamma \left(t\right),\;t\in \left(0,T_{e}\right],$$
with an initial condition of $u\left(t,x\right)=u_{0}\left(x\right)$. Here, $\frac{d}{dt}u$ is a material derivative, i.e.,
$$\frac{d}{dt}u=u_{t}+v\cdot \nabla u,$$
v is a velocity of the surface Γ(t) and *k* is a diffusion coefficient.

Let n denote the outward normal vector of the surface Γ(t). We assume that the velocity v contains both normal and tangential components. We denote by $v=Vn+v_{S}$ (with $v_{S}\cdot n=0)$ the velocity of material points on the surface. The tangential velocity $v_{S}$ transports *u* along Γ(t).

For a given time *t*, we denote by P the projection operator that projects a vector in $R^{3}$ at the point x into the $T_{x}\Gamma \left(t\right)$. Then P can be written as
$$P=I-nn^{T}.$$

Thus, we can define the surface gradient and Laplacian on Γ(t):$$\nabla_{\Gamma (t)}:=P\nabla ,\;\Delta_{\Gamma (t)}:=\nabla_{\Gamma (t)}\cdot \nabla_{\Gamma (t)}.$$

We represent the evolving surface Γ(t) by a parameterization approach. The velocity v determines the movement of nodes on Γ(t) by the following form:(2)$$\frac{d}{dt}x\left(t\right)=v\left(t,x\left(t\right)\right).$$
If (Equation 2) can be solved analytically, then we can define the parameterization of the Γ(t) over the initial surface Γ(0):Γ(t)={x(t)|x(0)∈Γ(0)}.
If not, we use the second order method [20] to numerically solve the ODE system (Equation 2):(3)$$x^{m+1}=x^{m}+\frac{\Delta t}{2}\left(3v^{m}-v^{m-1}\right).$$

## 3. ARBF-FD TPM for the Convection–Diffusion Equation on Evolving Surfaces

### 3.1. Anisotropic Radial Basis Function Interpolation

In this paper, we use a generalization form of RBF: ARBF. The purpose of using this basis function is that we want to benefit from its ability to handle the anisotropic growth of surfaces.

ARBF can be better understood from the two aspects: one is the RBF interpolant on transformed data, and the other is that using a new distance in the RBF. Let $X={\left\{x_{i}\right\}}_{i=1}^{N}\subset R^{d}$ denote an interpolated nodes set, where *d* is the dimension of space. We change the *X* into another data set $\tilde{X}$ through a linear transformation or a nonsingular d×d matrix *M*:$$\tilde{X}=\left\{\tilde{x_{i}}\;|\;\tilde{x_{i}}=Mx_{i},\;i=1,2,\cdots ,N\right\}.$$
Let $T=M^{T}M$, then *T* is symmetric positive definite. Results from [24] show that the RBF interpolation on $\tilde{X}$ is equivalent to the ARBF interpolation on *X*. The corresponding new norm is defined as
$${∥x∥}_{T}=\sqrt{x^{T}Tx}.$$

Now we can define the ARBF interpolant of a target function s(x):(4)$$I_{\phi }s\left(x\right)=\sum_{i=1}^{N}\alpha_{i}\phi \left({∥x-x_{i}∥}_{T}\right)+\sum_{k=1}^{m_{p}}\beta_{k}p_{k}\left(Mx\right).$$
It should be noted that the constraints on multivariate polynomials in (Equation 4) are
$$\sum_{k=1}^{N}\alpha_{k}p_{j}\left(Mx_{k}\right)=0,\;j=1,2,3,\cdots ,m_{p},$$
here, mp=l+dl and *l* is the largest degree of the polynomials. The coefficients $\alpha ={\left(\alpha_{1},\alpha_{2},\cdots ,\alpha_{N}\right)}^{T},\beta ={\left(\beta_{1},\beta_{2},\cdots ,\beta_{m_{p}}\right)}^{T}$ in (Equation 4) are obtained by solving the following linear system:$$\left(\begin{array}{cc} A & P\\ P^{T} & 0 \end{array}\right)\left(\begin{array}{c} \alpha \\ \beta \end{array}\right)=\left(\begin{array}{c} s_{X}\\ 0 \end{array}\right),$$
where $A=\left\{a_{ij}=\phi \left({∥x_{i}-x_{j}∥}_{T}\right)\right\},s_{X}={\left(s\left(x_{1}\right),s\left(x_{2}\right),\cdots ,s\left(x_{N}\right)\right)}^{T}$, and
$$P=\left(\begin{array}{cccc} p_{1}\left(Mx_{1}\right) & p_{2}\left(Mx_{1}\right) & \cdots & p_{m_{p}}\left(Mx_{1}\right)\\ p_{1}\left(Mx_{2}\right) & p_{2}\left(Mx_{2}\right) & \cdots & p_{m_{p}}\left(Mx_{2}\right)\\ \vdots & \vdots & \vdots & \vdots \\ p_{1}\left(Mx_{N}\right) & p_{2}\left(Mx_{N}\right) & \cdots & p_{m_{p}}\left(Mx_{N}\right) \end{array}\right).$$

In numerical computations, the behaviors of RBF interpolations are usually related to the separation distance and fill distance:(5)$$q_{X}=\frac{1}{2}\min_{1\leq i<j\leq N}∥x_{i}-x_{j}{∥}_{2},\;h_{X}=\underset{x\in \Omega }{\sup}\min_{1\leq i\leq N}{∥x-x_{i}∥}_{2},$$
where $\Omega \subset R^{d}$ is a computational domain. The condition number of the RBF interpolation matrix increases with the decrease in the $q_{X}$. The fill distance $h_{X}$ determines the accuracy of the interpolation. The two quantities of a near-uniform nodes set are close to each other.

When *M* (or *T*) is an identity matrix, the above ARBF interpolation turns into the RBF interpolation. If the surface does not have an anisotropic structure, the Euclidean distance is sufficient to give good results for the interpolation. For surfaces with a global anisotropic structure, such as ellipsoids, we need to select a matrix *M* that matches the anisotropic surface geometry. In other words, our goal is to seek a suitable matrix *M* such that the $h_{X}$ and $q_{X}$ of the mapped data $\tilde{X}$ are closer together. In this paper, the basis function is chosen as the anisotropic version of the Gaussian RBF:$$\phi \left({∥x-x_{i}∥}_{T}\right)=\exp\left(-\epsilon^{2}{∥x-x_{i}∥}_{T}^{2}\right),$$
where ϵ is a free shape parameter.

We know that the shape parameter also has an effect on the RBF interpolation. For a fixed node set, it leads to ill-conditioned problems as the shape parameter ϵ becomes small. In this case, RBF becomes flatter, and the local interpolation matrix causes an ill-conditioned linear system. The RBF-QR method [25,26] has been proposed to circumvent this problem. However, our study focuses on the influence of nodes distribution on the RBF interpolation. Therefore, instead of using the RBF-QR method, a suitable shape parameter is selected for each numerical test.

### 3.2. Differentiation Weights Based on TPM

In this section, we illustrate how to calculate the surface gradient and Laplacian using the ARBF-FD TPM. Without loss of generality, let $S_{1}={\left\{x_{j}\right\}}_{j=1}^{n}\subset X$ denote a stencil, where $x_{j},j=1,2,\cdots ,n$ are the nearest neighbors of the center $x_{1}$. Let *L* be a differential operator. The approximation of the Lu(x) at $x_{1}$ can be computed by the following linear combination:(6)$${Lu\left(x\right)|}_{x=x_{1}}\approx L_{X}^{1}u\left(x\right)=\sum_{j=1}^{n}\omega_{j}^{L}u\left(x_{j}\right).$$
Thus the weights $\omega_{j}^{L}$ are found by solving the system,
(7)$$\left(\begin{array}{cc} A & P\\ P^{T} & 0 \end{array}\right)\left(\begin{array}{c} \omega^{L}\\ \eta \end{array}\right)=\left(\begin{array}{c} {L\Phi \left(x\right)|}_{x=x_{1}}\\ {Lp\left(Mx\right)|}_{x=x_{1}} \end{array}\right),$$
where $\Phi (x)={\left(\phi \left({∥x-x_{1}∥}_{T}\right),\phi \left({∥x-x_{2}∥}_{T}\right),\cdots ,\phi \left({∥x-x_{n}∥}_{T}\right)\right)}^{T}$ and $p\left(Mx\right)={\left(p_{1}\left(Mx\right),p_{2}\left(Mx\right),\cdots ,p_{m_{p}}\left(Mx\right)\right)}^{T}$.

A straightforward way is that calculating the weights directly by (Equation 7) using a closed-form formulation of surface operators, as proposed in [27,28] (the method is called the RBF-FD direct method hereafter). In this paper, the process of calculating the weights is implemented within the TPM framework. The advantage of using this method lies in the fact that the proposed method does not require a closed-form formulation of surface operators. Thus, it is much easier to implement.

We describe this method in detail below. Let $\bar{u}$ denote the normal extension of *u*, i.e., $n\cdot \nabla \bar{u}=0$. The key idea of TPM is that the problem of computing the surface operator of *u* is reduced to that of computing the gradient (in $R^{3}$) of the $\bar{u}$. This is due to the fact that
$$\nabla_{\Gamma }u=\nabla \bar{u}-\left(n\cdot \nabla \bar{u}\right)n=\nabla \bar{u}.$$
Notice again that the gradient of $\bar{u}$ in the local orthonormal coordinate system $\left\{t_{1},t_{2},n\right\}$ can be written as
$$\nabla \bar{u}=\left(t_{1}\cdot \nabla \bar{u}\right)t_{1}+\left(t_{2}\cdot \nabla \bar{u}\right)t_{2}+\left(n\cdot \nabla \bar{u}\right)n=\left(t_{1}\cdot \nabla \bar{u}\right)t_{1}+\left(t_{2}\cdot \nabla \bar{u}\right)t_{2}.$$
Thus, we only need to compute the $t_{1}\cdot \nabla \bar{u}$ and $t_{2}\cdot \nabla \bar{u}$. Here, $t_{1}$ and $t_{2}$ are the orthogonal basis of the tangent plane $T_{x_{1}}\Gamma$. Further, the weights of the surface gradient are computed by
(8)$$\omega_{j}^{\nabla_{\Gamma }}=t_{1}\omega_{j}^{\nabla_{t_{1}}}+t_{2}\omega_{j}^{\nabla_{t_{2}}},\;\;j=1,2,\cdots ,n.$$
The above differentiation weights $\omega_{j}^{\nabla_{t_{1}}}$ and $\omega_{j}^{\nabla_{t_{2}}}$ correspond to the directional derivatives of a function in the directions of $t_{1}$ and $t_{2}$, respectively.

For computational simplicity, we project the stencil onto the tangent plane at the center $x_{1}$. Then we use a rotation $R={\left(t_{1},\;t_{2}\;,n\right)}^{T}$ to transform the projected nodes so that they are now located around a plane that is parallel to the plane z=0. Through the coordinate transformation
(9)$$x_{R}=R\;x,$$
we can obtain
$$\frac{\partial \bar{u}}{\partial x_{R}}=t_{1}\cdot \nabla \bar{u},\;\frac{\partial \bar{u}}{\partial y_{R}}=t_{2}\cdot \nabla \bar{u},$$
where $x_{R}$ and $y_{R}$ are the first and second components of $x_{R}$. Then we ignore the z-coordinates of $x_{R}$, construct 2D RBFs on the rotated stencil and compute the weights corresponding to $\frac{\partial \bar{u}}{\partial x_{R}}$ and $\frac{\partial \bar{u}}{\partial y_{R}}$.

To sum up the above, the calculation of the differentiation weights of the surface gradient requires two steps. The first step is to calculate the 2D weights on the rotated stencil by solving (Equation 7). The second step is to transform the weights to the original stencil by using (Equation 8).

For the discretization of the surface Laplacian, the RBF-FD TPM only needs the normal vectors. However, the RBF-FD direct method requires reconstructing the curvatures of surfaces. In addition, the RBF-FD TPM uses fewer terms in the polynomial basis in comparison to the RBF-FD direct method. This is because the 2D basis functions are used in the RBF-FD TPM.

We found that the transport scheme on sphere [29] is also essentially a tangent plane method, which uses Householder reflection to implement the process of rotation. However, the Householder reflection can only handle the surface of a sphere. For general surfaces, it is necessary to find $t_{1}$ and $t_{2}$ and construct the matrix *R*. If Γ is a parametric surface defined by x=x(ψ,θ), then $t_{1}$ and $t_{2}$ can be obtained directly by normalizing $x_{\psi }$ and $x_{\theta }$.

For the calculation of the surface Laplacian, we use the rotational invariance of the Laplacian or the proof in [19]. Therefore, we can obtain
$$\omega_{j}^{\Delta_{\Gamma }}=\omega_{j}^{\Delta_{T}},\;\;j=1,2,\cdots ,n,$$
where $\Delta_{T}$ is the 2D Laplacian on $T_{x_{1}}\Gamma$.

## 4. Discretization in Time

The Lagrangian framework is considered for the numerical discretization of (Equation 1). Let time steps $t^{m}=m\Delta t,\;m=0,1,\cdots ,M_{e},\Delta t=T_{e}/M_{e}$. At each time layer, we update the nodes $x^{m}$ through (Equation 3). Then we calculate the required differentiation weights for each center node by repeating the procedure presented in Section 3.2. Further, the differentiation matrix $D^{m}$ is constructed by the differentiation weights. Differentiation weights can be computed in parallel, as the procedure for each point is independent of that for all other points.

The material derivative of *u* can be simply approximated as
$$\frac{d}{dt}{u|}_{\left(t^{m},x^{m}\right)}\approx \frac{u\left(t^{m},x^{m}\right)-u\left(t^{m-1},x^{m-1}\right)}{\Delta t}.$$
We use the Crank–Nicolson scheme for time discretization. The fully discrete form is written as
(10)$$\left(I-\frac{\Delta t}{2}D^{m}\right)U^{m}=\left(I+\frac{\Delta t}{2}D^{m-1}\right)U^{m-1}+\frac{\Delta t}{2}\left(F^{m}+F^{m-1}\right),\;m=1,2,\cdots ,M_{e},$$
where $U_{i}^{m}$ is the approximated value of $u(t^{m},x_{i}\left(t^{m}\right))$ and $F_{i}^{m}=f\left(t^{m},x_{i}\left(t^{m}\right)\right)$.

## 5. Differentiation Accuracy

For the computation of PDEs on stationary or moving surfaces, calculating accurate differentiation weights turned out to be essential in the RBF-FD context. In this section, we show the errors and convergence orders of the proposed method for the approximation of the surface Laplacian. This test is conducted over the ellipsoid given by
(11)$$\Gamma =\{x\;|\;\frac{x^{2}}{9}+\frac{y^{2}}{9}+z^{2}=1\}.$$
We consider the function
(12)u(x,y,z)=sin(x) sin(y) sin(z)+cos(x) cos(y) cos(z).
The error is measured by the following form:$${\mathrm{Error}}_{\Delta_{\Gamma }}=\max_{1\leq i\leq N}\left|\Delta_{\Gamma }u\left(x_{i}\right)-L_{\Delta_{\Gamma },X}^{i}u\left(x\right)\right|,$$
where $L_{\Delta_{\Gamma },X}^{i}$ represents the approximation given by (Equation 6) for the surface Laplacian. For the surface, we map ’minimal energy’ (ME) nodes on the unit sphere to the ellipsoid (Equation 11). This node layout is illustrated in Figure 1.

### 5.1. On the Choice of the Matrix M

It is worth pointing out that the using of a suitable *M* matrix will improve the accuracy and well-conditioning of the interpolation problem (Equation 4). Therefore, it leads to better results for the computation of differentiation weights. Note again that the nodes on the ellipsoid are obtained by mapping the ME nodes on the unit sphere. An additional point to note is that the RBFs used in TPM are the 2D basis functions defined on rotated stencils. Therefore, for each stencil, we define a matrix $M_{2}$:(13)$$M_{2}=R_{s}M_{1}R_{e}^{-1},$$
where $M_{1}=diag\left(\frac{1}{3},\frac{1}{3},1\right),R_{s}={\left(t_{1}^{s},\;t_{2}^{s},\;n^{s}\right)}^{T},R_{e}={\left(t_{1}^{e},\;t_{2}^{e},\;n^{e}\right)}^{T}$. Here, $R_{s}$ and $R_{e}$ consist of the tangent vectors and normal vectors of the unit sphere and the ellipsoid at the center of the stencil, respectively. The role of $M_{2}$ is that of the nodes mapped to the unit sphere before rotating through (Equation 9) and then performing the rotation step by $R_{s}$.

In the implementation, $M_{2}$ is only used to construct a 2 × 2 matrix *M* by extracting the first two rows and columns of $M_{2}$. We then define the 2D ARBFs using the matrix *M* and compute differentiation weights using (Equation 7). Additionally, we construct an anisotropic type of stencils. That is, the distance between the center and nearest neighbors is measured using the ${∥\cdot ∥}_{T_{1}}$, where $T_{1}=M_{1}^{T}M_{1}$.

### 5.2. Results on the Ellipsoid

For the choice of parameters, we first choose the degree of polynomials with l=1,2 and 3. Then, we have set $n=2m_{p}+1$, where mp=l+3l and fix ϵ=3. To explain that the ARBFs would work better than RBFs, we compare the accuracy of the RBF-FD TPM and ARBF-FD TPM. Several observations can be made in Figure 2.


The errors of the RBF-FD TPM are relatively large. The reason for this is that the irregularity of nodes leads to poor RBF interpolation. The irregularity of nodes is reflected by the difference between the fill distance and separation distance. For convenience, the fill distance in (Equation 5) is approximated by
$${\bar{h}}_{X}=\frac{1}{2}\max_{1\leq j\leq N}\min_{1\leq i\leq N,i\neq j}{∥x_{j}-x_{i}∥}_{2}.$$
For 6561 nodes on the ellipsoid, ${\bar{h}}_{X}=0.0697$ and $q_{X}=0.0214$, the difference between the two quantities is relatively large;The ARBF-FD TPM has smaller errors and faster convergence rates than those of RBF-FD TPM. The $M_{1}$-matrix maps the nodes on the ellipsoid to the near-uniform ME nodes on the unit sphere. The corresponding two distances of the transformed nodes set are ${\bar{h}}_{X}=0.0235$ and $q_{X}=0.0210$. We obtained better results by introducing a new distance to reduce the difference between these two quantities. The convergence orders are 1.43, 3.67 and 6.19 for different *l*, respectively.


In Figure 3, we give the eigenvalues of the RBF-FD TPM and ARBF-FD TPM corresponding to the surface Laplacian differentiation matrix on the ellipsoid, where l=2, n=21, ϵ=3 and N=6561. Due to the irregular sampling of the nodes on the ellipsoid, the RBF-FD TPM method has large positive eigenvalues. In the ARBF-FD TPM method, we improved this problem using a new metric. Therefore, the eigenvalues of the ARBF-FD TPM are all in the negative half-plane.

For this test, there are two other ways to improve the results of the RBF-FD TPM. The first one is to use the method proposed in [12], fixing the condition number of local interpolation matrix and choosing stencil-dependent shape parameters. Compared with the ARBF method, choosing different shape parameters in each stencil can be regarded as the isotropic scaling of each stencil. For the evolving surface problems, this strategy increases the computational cost. The second one is to sample near-uniform nodes on the ellipsoid. This can improve the stability of eigenvalues for the ellipsoid. However, if we consider the anisotropic growth of the unit sphere, regenerating the near-uniform nodes at each time layer would be time consuming.

Summarizing the above discussion, the use of the ARBFs overcomes the ill-conditioned interpolation problem caused by irregular sampling of the nodes without increase of the computation cost. This means that the ARBF-FD TPM method is effective for solving the Poisson or diffusion equations on the stationary ellipsoid. In this paper, we mainly focus on the application of the ARBF-FD TPM for the anisotropic growth of surfaces.

## 6. Numerical Experiments

### 6.1. Normal and Tangential Motion on the Unit Sphere

First, we test the accuracy of the proposed method for the Equation (Equation 1) on surfaces without anisotropy. In this case, we suggest using isotropic RBFs (i.e., *M* is the identity) for the numerical simulations. For this, we consider two examples: an expanding sphere and a rotating sphere. The velocity of the expanding sphere has only a normal component, while the velocity of rotating sphere is purely tangential.

#### 6.1.1. Expanding Sphere

For the first example, we consider the expanding sphere,
$$\Gamma \left(t\right)=\{x\left(t\right)\;|\;x^{2}+y^{2}+z^{2}={\left(1+0.5t\right)}^{2}\}.$$
Then the velocity is v=0.5n. In Equation (Equation 1), the diffusion coefficient is chosen to be k=1. We assume that the exact solution is given by
$$u\left(t,x\right)=e^{-6t}xy,$$
The time interval of the simulation is set to be (0,1].

At the initial time, we use ME nodes on the unit sphere. The values of the parameters are given in Table 1. For the two cases listed in the table, we choose $\Delta t={\bar{h}}_{X}/5$ and $\Delta t={\bar{h}}_{X}^{2}/30$, respectively. The relative $l_{2}$ error in the numerical solution is measured at $T_{e}=1$ by
$$l_{2}\;\mathrm{error}={\left(\frac{\sum_{i=1}^{N}{\left(U_{i}^{M_{e}}-u\left(T_{e},x_{i}\left(T_{e}\right)\right)\right)}^{2}}{\sum_{i=1}^{N}u{\left(T_{e},x_{i}\left(T_{e}\right)\right)}^{2}}\right)}^{1/2}.$$

The errors of the RBF-FD TPM for the test are given in Figure 4. Second- and fourth-order convergence are obtained for the cases of ① and ②, respectively. It can be found that the nodes remain near-uniformly distributed as the expanding of the sphere. That is, $h_{X}$ and $q_{X}$ are relatively close to each other during the surface evolution. For this case, we can see from Figure 4 that higher convergence orders are obtained by the using of the Euclidean distance in RBFs.

#### 6.1.2. Rotating Sphere

In this example, we choose a velocity of $v={\left(y,\;-x,\;0\right)}^{T}$, which is a divergence free tangential velocity. Therefore, Equation (Equation 1) can be simplified as
$$u_{t}+v\cdot \nabla_{\Gamma }u-k\Delta_{\Gamma }u=f.$$
This equation can also be viewed as a convection–diffusion equation on the stationary unit sphere and can be numerically solved in the Eulerian framework. For pure convection or convection-dominated cases, the Eulerian RBF-FD method usually has spurious oscillations. This needs to rectified with artificial hyperviscosity [15]. The fully Lagrangian method used in this paper does not require the artificial hyperviscosity.

For this example, let f=0 and $k=10^{-5}$. The initial condition is taken as a cosine bell:(14)$$u_{0}\left(x\right)=\left\{\begin{array}{cc} \frac{1}{c}\left(1+\cos(\frac{\pi \;\arccos\;x}{1/3})\right), & \;\arccos\;x\;<\;\frac{1}{3},\\ 0, & \;\arccos\;x\;\geq \;\frac{1}{3}, \end{array}\right.$$
where c=2. We simulate the numerical solutions in Figure 5 at times t=0,2π and 20π. Here we using the parameters of N=6561,Δt=0.005 and ① given in Table 1. There are no spurious oscillations in the numerical solution because the method does not require the discretization of the surface gradient. Numerical results demonstrated the validity of the RBF-FD TPM for the convection-dominated convection–diffusion equations.

### 6.2. Evolving Ellipsoid

In this example, we wanted to present the effectiveness of the ARBF-FD TPM method for the anisotropic evolution of surfaces. We consider Equation (Equation 1) on an evolving ellipsoid:$$\Gamma \left(t\right)=\{x\left(t\right)\;|\;\frac{x^{2}}{a{\left(t\right)}^{2}}+\frac{y^{2}}{b{\left(t\right)}^{2}}+\frac{z^{2}}{c{\left(t\right)}^{2}}=1\},$$
where a(t)=1+4t,b(t)=1+2t and c(t)=1+t. We assume that the nodes move in the following way:x(t)=a(t)x(0),  y(t)=b(t)y(0)  and  z(t)=c(t)z(0).
Then the velocity of the surface is computed by $v\left(t,x\right)={\left(\frac{a^{\prime }\left(t\right)}{a(t)}x,\;\frac{b^{\prime }\left(t\right)}{b(t)}y,\;\frac{c^{\prime }\left(t\right)}{c(t)}z\right)}^{T}$.

We assume that the exact solution is given by
$$u\left(t,x\right)=e^{-t}\sin\left(x\right),$$
and the diffusion coefficient k=1. The final time is taken to be $T_{e}=1$. The final surface is an ellipsoid which has different growth rates in the x, y and z directions.

The velocity v leads to an irregular clustering of nodes with the increase in time. In Figure 6, we show how the distribution of nodes changes with the increase in time. ${\bar{h}}_{X}$ and $q_{X}$ corresponding to the nodes set in each subgraph in Figure 6 are given in Table 2. With the anisotropic growth of the surface, these two quantities display differences between themselves.

The new metric in ARBFs is constructed in the same way as described in Section 5.1. At a given time *t*, the $M_{1}$-matrix in (Equation 13) is chosen by
$$M_{1}\left(t\right)=diag\left(\frac{1}{a(t)},\frac{1}{b(t)},\frac{1}{c(t)}\right).$$
The matrix $M_{1}$ maps the nodes to the initial surface (the unit sphere) at each time layer. In more detail, the $h_{X}$ and $q_{X}$ of the mapped data are closer together, which will lead to more stable and accurate results. If we consider a more general case of the surface evolution,
x(t)=B(t)x(0),
then we can choose $M_{1}\left(t\right)=B{\left(t\right)}^{-1}$ for this case.

Use the same parameters as the test of the expanding sphere, given in Table 1. For ① and ②, the time step is chosen as $\Delta t={\bar{h}}_{X}/5$ and $\Delta t={\bar{h}}_{X}^{2}$, respectively. In Figure 7, we give the errors of the RBF-FD TPM and ARBF-FD TPM. For the anisotropic surface evolution, the Euclidean distance is not appropriate. As a result, the RBF-FD TPM method shows eigenvalue instability. However, the proposed ARBF-FD TPM method does not suffer from the irregular clustering of nodes. This is because we map the nodes to the initial surface at each time layer. Further, the values of ${\bar{h}}_{X}$ and $q_{X}$ of the transformed data stay the same as the initial moment. The advantage of the method is that it not only overcomes the instability caused by the irregularity of nodes, but also avoids the loss of accuracy caused by the growth of ${\bar{h}}_{X}$.

### 6.3. Evolving Torus

We consider the anisotropic evolution on the torus:(15)$$\Gamma \left(t\right)=\{x\;|\;{\left(x^{2}+y^{2}+{\left(1+2t\right)}^{2}z^{2}+0.84\right)}^{2}-4\left(x^{2}+y^{2}\right)=0\}.$$
Thus, the position of the nodes is given by $x\left(t\right)={\left(x\left(0\right),\;y\left(0\right),\;\frac{z(0)}{1+2t}\right)}^{T}$. This surface becomes very thin as it shrinks in the z-direction. It can be seen from Table 3 that the separation distance decreases with time, which leads to an ill-conditioned RBF interpolation. Following the idea of the previous example, the $M_{1}$-matrix is chosen by $M_{1}\left(t\right)=diag\left(1,1,1+2t\right)$. The matrix transforms the nodes at time *t* to the initial surface. Then, we replace $R_{s}$ and $R_{e}$ in (Equation 13) with the rotation matrix corresponding to the torus at the initial time and time *t*.

We simulate the diffusion on the evolving torus with $k=10^{-1}$. Initial condition is chosen as the cosine bell (Equation 14) with c=800. Figure 8 shows the evolution of the surface and the numerical solutions at times t=0, t=1 and t=2. We use 6230 near-uniform nodes on the initial surface and ② given in Table 1. For the test, let Δt=0.001. This surface evolution is anisotropic shrinking, as reflected by the fact that $q_{X}$ decreases much faster than ${\bar{h}}_{X}$. The mapped nodes (i.e., nodes at t=0) present increased $q_{X}$, with consequent reduction of the condition number of the RBF interpolation matrix. This gives a better quality of the ARBF interpolant. Thus, the ARBF-FD TPM method accurately simulates the problem, even when the torus becomes very thin.

### 6.4. Solid Tumor Growth

In more general evolutions of surfaces, the velocity is often coupled with the solution of the PDE on the surfaces, such as the model of solid tumor growth [3,4,16,17]. This model is mathematically described by the following reaction–diffusion system [16]:(16)$$\begin{array}{cc} \; & \frac{d}{dt}u+u\nabla_{\Gamma (t)}\cdot v-\Delta_{\Gamma (t)}u=f_{1}\left(u,w\right),\\ \; & \frac{d}{dt}w+w\nabla_{\Gamma (t)}\cdot v-d_{c}\Delta_{\Gamma (t)}w=f_{2}\left(u,w\right), \end{array}$$
where
$$f_{1}\left(u,w\right)=\gamma \left(a-u+u^{2}w\right),\;\;\;\;and\;\;\;\;\;f_{2}\left(u,w\right)=\gamma \left(b-u^{2}w\right).$$

For the system, we choose $d_{c}=10,\gamma =200,a=0.1$ and b=0.9. The velocity is chosen to be
$$v=\left(-0.01\kappa +0.4u\right)n+v_{a},$$
where $\kappa =\nabla_{\Gamma }\cdot n$ denotes the mean curvature, and $v_{a}$ is a given anisotropic velocity. We observed in the previous articles that the model does not have $v_{a}$. Note that the velocity v is no longer purely in the normal direction, but includes a tangential component. We compute the surface divergence of the velocity by
$$\nabla_{\Gamma (t)}\cdot v=\left(-0.01\kappa +0.4u\right)\kappa +\nabla_{\Gamma (t)}\cdot v_{a}.$$

For the test, use N=6561 nodes on the unit sphere and N=6230 nodes on the torus Γ(0) given by (Equation 15) as initial data. The initial conditions are obtained from the simulation of the model (Equation 16) in the stationary case (i.e., v=0) for these two surfaces at t=1.7 and t=1, respectively. In the scheme (Equation 10), the nonlinear source term is computed by the following form
$$\frac{1}{2}\left(3f_{i}\left(U^{m-1},W^{m-1}\right)-f_{i}\left(U^{m-2},W^{m-2}\right)\right),\;i=1,2.$$

We choose $v_{a}={\left(\frac{4}{1+4t}x,\;\frac{2}{1+2t}y,\;\frac{1}{1+t}z\right)}^{T}$ for the evolution of the sphere and $v_{a}=\left(0,\;0,\;0\right)$ for the torus, respectively. For the test on the sphere, we construct ARBFs in the way illustrated in Section 6.2. For the case on the torus, we use isotropic RBFs.

Note that at each time layer, the normal vector is required for numerical computation. Here, we compute a weighted normal, which is an average (corresponding to the area of each triangle) of all the triangle normals belonging to triangles with the given nodes.

Let Δt=0.001. Here, we use the cases of ② and ① (given in Table 1) for the tests on the sphere and the torus, respectively. Numerical approximations of *u* on the two surfaces are given in Figure 9 and Figure 10. It can be found that the surface expands outwards in regions where the solution *u* is large. In particular, solid tumor growth on the sphere becomes a bumpy ellipsoid. This is because it contains an artificial anisotropic velocity.

The example demonstrates that our method is effective for solving PDEs on moving point clouds. For this, we only need to calculate the normal vector through the triangular mesh. Although, the velocity v contains the curvature, we replace this term with the relation $\kappa n=-\Delta_{\Gamma }x$. Notice that the velocity also leads to a less uniform distribution of nodes, but the changes of ${\bar{h}}_{X}$ and $q_{X}$ are not significant during the time interval we tested. Some adaptive mesh refinement strategy may be needed if we want to simulate long time evolution.

## 7. Conclusions

In this work, we developed an RBF-FD method that can be employed for practical applications of PDEs on evolving surfaces. This proposed method is based on the tangent plane approach for calculating differentiation weights. The most attractive advantage of the method is that it transforms the problem into computing weights in 2D Euclidean space.

To make the RBF-FD TPM more efficient for anisotropic surface evolutions, we propose the new ARBF-FD TPM. The proposed method gives a significant improvement in numerical experiments. To construct the ARBFs, we define a linear mapping such that the nodes at each time layer are mapped to the initial surface. This approach gives advantages because it is possible to reduce the impact on the irregular clustering of nodes during the anisotropic surface evolution. An extension of the ARBF-FD TPM would be to adapt the method to handle anisotropic diffusion on surfaces. Our method can obtain higher convergence order by increasing the stencil size and degree of polynomials. Furthermore, the proposed ARBF-FD TPM can be applied directly to PDEs on evolving surfaces represented by triangular meshes or point clouds. Therefore, the method is potentially useful for computing PDEs on evolving surfaces.

A limitation of the ARBF-FD TPM is that the current results may be only applicable to the surface evolution in ways that depend linearly on the underlying axes. For evolving surface problems which are stretched in nonlinear ways, it may be necessary to find a nonlinear mapping. This approach requires defining a more complex basis function than ARBF. Moreover, we will extend the method to handle complex surfaces with large deformations, such as for the topological change of surfaces.

## Figures and Tables

**Figure 1 entropy-24-00857-f001:**
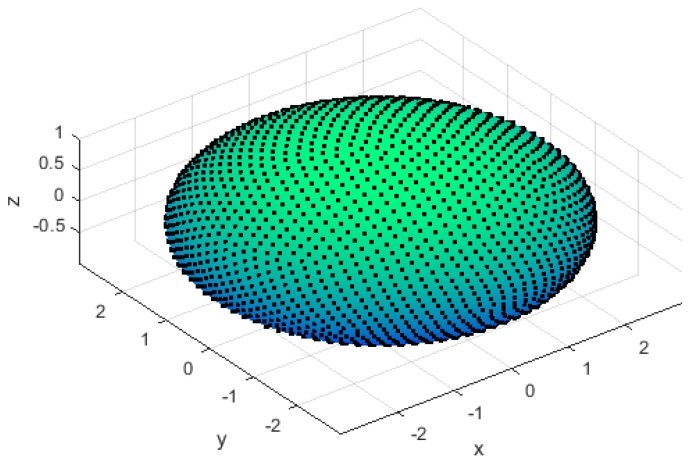
Node layout on the ellipsoid.

**Figure 2 entropy-24-00857-f002:**
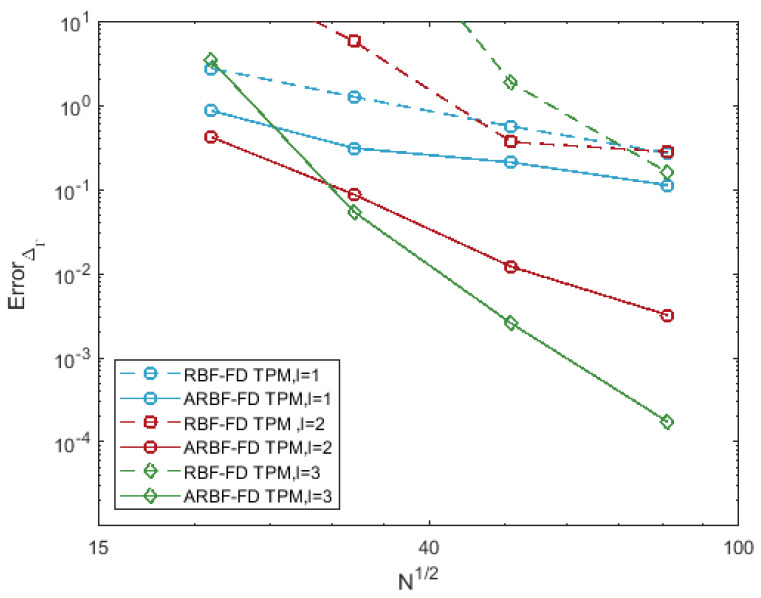
Errors for surface Laplacian of function (Equation 12) on the ellipsoid.

**Figure 3 entropy-24-00857-f003:**
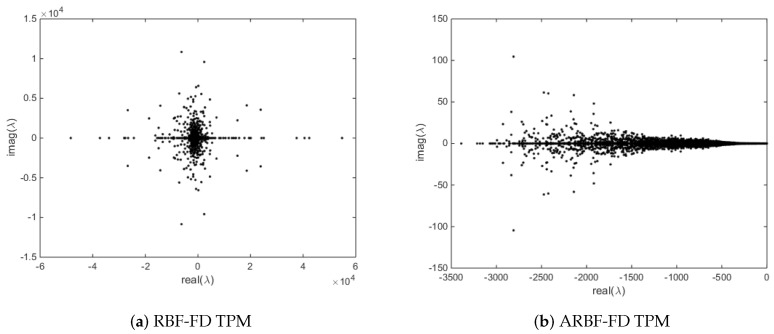
Eigenvalues of the two methods for the surface Laplacian on the ellipsoid.

**Figure 4 entropy-24-00857-f004:**
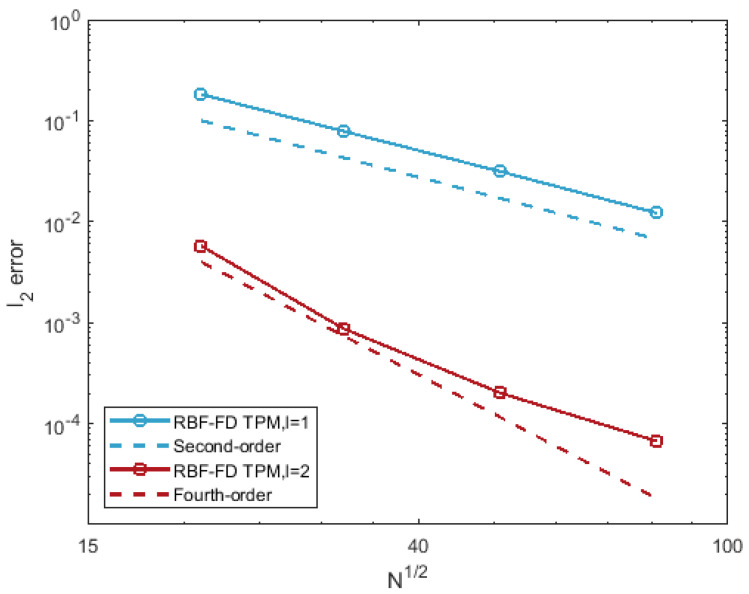
Relative $l_{2}$ error of the RBF-FD TPM for the test of expanding sphere at $T_{e}=1$.

**Figure 5 entropy-24-00857-f005:**
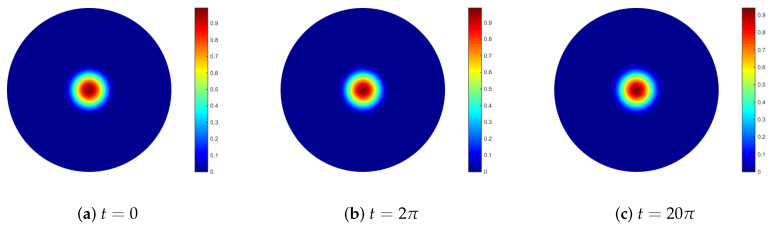
Numerical solution for the test of rotating sphere.

**Figure 6 entropy-24-00857-f006:**
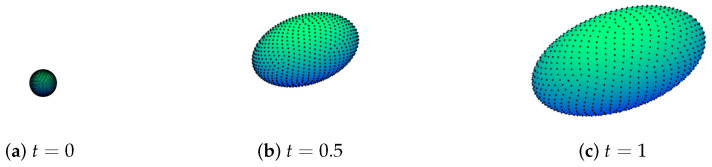
Distribution of nodes on the evolving ellipsoid, where N=1024.

**Figure 7 entropy-24-00857-f007:**
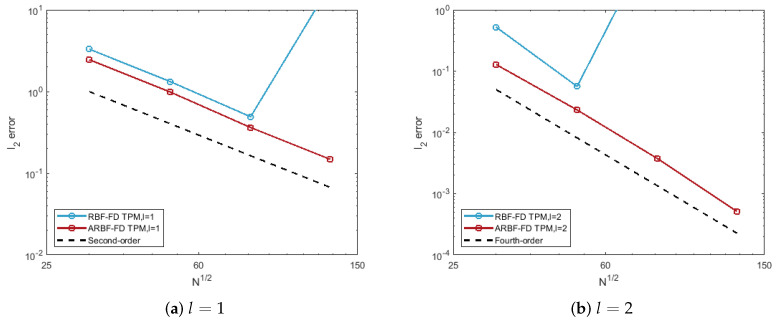
Relative $l_{2}$ errors of the RBF-FD TPM and ARBF-FD TPM for the test of evolving ellipsoid at $T_{e}=1$.

**Figure 8 entropy-24-00857-f008:**
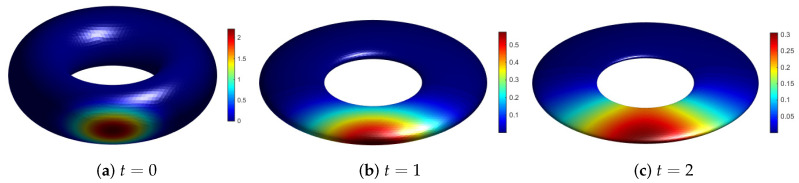
Numerical solution on the evolving torus.

**Figure 9 entropy-24-00857-f009:**
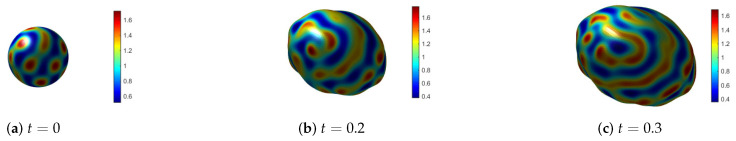
Numerical solution for the solid tumor growth on the sphere.

**Figure 10 entropy-24-00857-f010:**
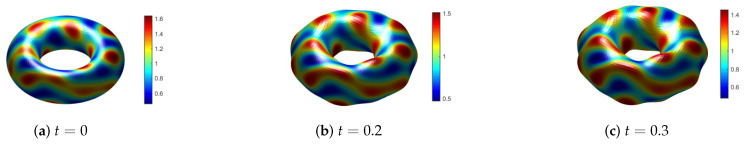
Numerical solution for the solid tumor growth on the torus.

**Table 1 entropy-24-00857-t001:** The values of the parameters.

Case	*l*	*n*	ϵ
①	1	9	1
②	2	21	2

**Table 2 entropy-24-00857-t002:** The fill and separation distance of the nodes on the ellipsoid, where N=1024.

Distance	t=0	t=0.5	t=1
${\bar{h}}_{X}$	5.9378 × 10${}^{-2}$	1.2999 × 10${}^{-1}$	2.0258 × 10${}^{-1}$
$q_{X}$	5.3421 × 10${}^{-2}$	8.5483 × 10${}^{-2}$	1.1430 × 10${}^{-1}$

**Table 3 entropy-24-00857-t003:** The fill and separation distance of the nodes on the torus, where N=6230.

Distance	t=0	t=1	t=2
${\bar{h}}_{X}$	2.8538 × 10${}^{-2}$	2.7508 × 10${}^{-2}$	2.7501 × 10${}^{-2}$
$q_{X}$	2.1260 × 10${}^{-2}$	7.8858 × 10${}^{-3}$	4.7380 × 10${}^{-3}$

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
