# Peer review of "Numerical Study on an RBF-FD Tangent Plane Based Method for Convection–Diffusion Equations on Anisotropic Evolving Surfaces"

_entropy, 2022, doi:10.3390/e24070857_

Round 1
Reviewer 1 Report
Dear authors,
Overall, it is a well-written manuscript. But at different parts of the manuscripts, there are long sentences that might be addressed by diving them into shorter sentences. That would also help the fluency of the manuscript.
The paper provides experimental results to validate the method. More in-depth discussion after experiments and implementation of the method would make it more clear the how strong the method is.
Please add more clearly discussion how the method is potentially useful in what sense for computing PDEs on evolving surfaces.
Reviewer 2 Report
The paper includes a number of numerical experiments with good results. I recommend the article for publication. However, I have few questions for authors: 1) Would it be important to know the stencil's sizes? 2) Using the direct method of RBF-FD frequently results in ill-conditioned problems. What do you think about this? 3) Can the method be used to solve nonlinear problems?
